# Effects of the Oxytocin Hormone on Pelvic Floor Muscles in Pregnant Rats

**DOI:** 10.3390/medicina59020234

**Published:** 2023-01-26

**Authors:** Emine Demir, Sukriye Deniz Mutluay, Hacer Sinem Buyuknacar

**Affiliations:** 1Department of Midwifery, Ege University Faculty of Health Sciences, 35575 Izmir, Turkey; 2Department of Midwifery, Cukurova University Faculty of Health Sciences, 01330 Adana, Turkey; 3Department of Pharmacology, Cukurova University Faculty of Pharmacy, 01330 Adana, Turkey

**Keywords:** birth, induced labor, oxytocin, pelvic floor, rats

## Abstract

*Background and Objectives:* Oxytocin induction is a known risk factor for pelvic floor disorders (PFDs). The aim of the study was to investigate the effects of oxytocin induction on pelvic floor muscles in pregnant rats. *Methods:* Thirty-two female Wistar rats were included and divided into four groups (n = 8). The groups were as follows: virgin group (group I)–from which muscles were dissected at the beginning of the experiment; spontaneous vaginal delivery (group II) which has delivery spontaneously; saline control group (group III) and oxytocin group (group IV). In groups III and IV, pregnancy was induced on d 21 of pregnancy, with 2.5 mU saline solution or iv oxytocin, respectively, delivered by the intravenous (iv) route in pulses at 10-min intervals for 8 h. Then, the rats were euthanized, the m. coccygeus, m. iliocaudalis and m. pubocaudalis muscles were excised and tissue samples were taken. After histological processing, the vertical and horizontal dimensions of the muscles were analyzed under a light microscope. *Results:* In group IV; the measurement of the horizontal dimension of the m. pubocaudalis muscles was 50.1 ± 5.4 µm and it was significantly higher than other groups (*p* < 0.001). In group III; the mean value of the horizontal dimension of m. coccygeus muscle was found to be 49.5 ± 10.9 µm and it was significantly higher than other groups (*p* < 0.009). Between-group comparisons revealed no difference in mean m. iliocaudalis muscle dimension (*p* > 0.05). *Conclusions:* As a result of our study it can say that whether oxytocin induced or not, vaginal birth is a process that affects the pelvic muscles.

## 1. Introduction

The pelvic floor muscles (PFMs) are related to the compound structure which closes the bony pelvic outlet and is a muscular layer of the pelvic floor which comprises the m(musculus). coccygeus and the m. levator ani complex and supports the pelvic floor function [1,2]. PFMs play an important role in supporting the pelvic organs [3]. They are also very important during coughing, sneezing, strong respiration, defecation and intra-abdominal pressure during birth [3,4,5,6]. In addition, these muscles ensure internal rotation of the presenting part of the fetus during delivery and its progression through the birth canal [3,7].

Pregnancy and childbirth are common risk factors for complications such as placenta previa, uterine rupture, preeclampsia and pelvic floor disorders (PFDs) [8,9,10]. The symptoms of PFDs include urinary incontinence, bladder storage and voiding and post-micturition, pelvic organ prolapse, sexual dysfunction, anorectal dysfunction, lower urinary tract pain and/or other pelvic pain and lower urinary tract infections [11,12,13,14,15,16,17,18,19]. These clinical manifestations of PFDs place a large financial burden on health systems [11,12,13,14,15,16,17,18,19]. In addition, they give rise to major social, physical and psychological problems and reduce the quality of life and reproductive ability of women [14,15,17,18,19,20,21].

Although PFDs are painful and have adverse effects on quality of life, most women do not seek health care because of a lack of knowledge, shame or fear of complex treatments [20,22]. Thus, it is difficult to determine the prevalence of PFDs [20]. Previous studies showed that 11–35.5% of women worldwide suffered from PFDs, with figures of 7.6–13%, 1.7–35.5%, 2.9–20% and 0.2–13% reported for overactive bladder, urinary incontinence, pelvic organ prolapse and anal incontinence, respectively [22].

The causes of PFDs are not well understood, but they can be classified according to their origin into obstetric and non-obstetric causes. Advanced age, obesity or having a high body mass index, as well as heavy lifting, smoking, excessive exercise, connective tissue disorders and neurological diseases, can be considered non-obstetric causes [8,10,14,15,19,20,23,24]. Obstetric causes include parity, operative vaginal delivery, duration of the first and second stages of labour, episiotomy, epidural analgesia, laparoscopic and hysteroscopic interventions, occiput posterior presentation, macrosomy, foetal head size, obstetric trauma, perineal laceration, prostaglandin and oxytocin induction [10,15,19,20,21,23,24,25,26].

Oxytocin hormone is the most widely used pharmacological agent in induction and augmentation of labor [27,28,29]. Along with its widespread use, in 2007 the Institute for Safe Medication Practices and the U. S. Food and Drug Administration (FDA) added intravenous oxytocin administration to the high-risk drug list [30,31]. It is still a great challenge to predict whether oxytocin, which is on a high-risk drug list, affects maternal and neonatal outcomes due to misuse or oxytocin itself [32]. Therefore, the consequences of induction and augmentation of labor with oxytocin, both positive and negative effects on maternal and fetal-neonatal health, need to be investigated [32].

It is important to determine the changes that occur in the pelvic floor muscles during pregnancy in order to understand the effects of oxytocin hormone on pelvic floor disorders. It has been reported in many studies that important biomechanical and biochemical changes occur in the pelvic floor muscles while preparing for delivery from the pregnancy process [2,8,11]. Imaging parameters of the levator hiatus are used to detect these changes. However, examining the dimensions of the levator hiatus provides very limited information about pelvic floor musculature, strength and contractile capacity. The fact that the changes that pregnancy and childbirth cause on the pelvic floor muscles are not fully known makes it impossible to develop protective strategies [2].

Structural studies require complete isolation of the muscle to be examined. Since it is not possible to carry out such studies on humans, experimental animals are used [2,18]. Studies show that the pelvic floor muscles of primates are the best model in terms of similarity to human structure [18]. However, since studies on primates are very expensive and logistically difficult, other alternatives must be used [18]. Rabbits, mice and rats, which are more convenient than primates in terms of cost and production, are commonly preferred in the examination of pelvic floor muscles. In many studies, it has been determined that the m. coccygeus, m. iliocaudalis and m. pubocaudalis muscles in the pelvic floor muscles of rats are the most anatomically similar to the human pelvic floor muscles [2]. In addition, pregnancy-related changes in the cervix, vagina and supporting structures of the vagina of rats are very similar to the changes that occur in human anatomy [2].

Oxytocin induction is an accepted risk factor for PFDs. However, there are only a few studies on this subject, and no consensus on its role in PFDs has been reached [33,34,35]. Therefore, in this study we investigated the effects of oxytocin induction on PFMs in pregnant rats.

## 2. Materials and Methods

This was a randomized controlled trial. Since Clinical.trials only records human subject studies, the study could not be registered. The Cukurova University Animal Experiments local ethics committee approved all procedures (4 July 2018 with decision number 21). The experiments were performed between 15 September 2018 and 15 May 2019. All experiments were performed in accordance with guidelines on the care and use of animals.

The animals were housed-caged at an average ambient temperature of 21 ± 2 °C under a 12 h light/12 h dark cycle and food and water were provided ad libitum.

In total, 32 female Wistar Albino rats with a mean weight of 150–200 g were randomly divided into four groups (n = 8/group).

Group I: Virgin rats which had not given delivery before.

Group II: Pregnant rats which had spontaneous vaginal delivery.

Group III: Rats induced with saline solution intravenously (iv) on d 21 of pregnancy at 10 min intervals for 8 h and serving as a control group.

Group IV: Rats induced with oxytocin solution (iv) on d 21 of pregnancy at 10 min intervals for 8 h and serving as the experimental group.

At the age of 3 mo, the rats in group I (virgin group) were anesthetized under deep anesthesia, and the PFMs were dissected. All the rats in the other three delivery groups (groups II, III and IV) reached 3 mo of age, and oestrus cycles were checked daily by a vaginal smear. Upon estrus detection, each female rat was placed in a cage with four males. The day of vaginal plug development was accepted as the first day of pregnancy, and the male rats were removed from the cages. The rats in group II were anesthetized under deep anesthesia after spontaneous vaginal delivery, and the PFMs were dissected.

On d 21 of pregnancy, each rat was weighed, and Xylazine (%2 Rompun, Bayer; 10 mg/kg) and Ketamine (Ketalar, Parke-Davis, Sydney, Australia; 80 mg/kg) were administered intraperitoneally to provide deep anesthesia. After deep anesthesia, the rats were laid on their back, and iv catheters were inserted into the tail area. After catheterization, the rats in group III (saline solution group) received saline pulses every 10 min for 8 h according to their body weight. The rats in group IV received pulses of oxytocin solution (Synpitan Forte 5 IU/mL) every 10 min for 8 h according to their body weight. At the end of the 8-h experimental period, the rats in group III (saline solution group) and group IV (oxytocin group) were returned to their cages to continue with their pregnancies. After the rats (group III and IV) were returned to their cages, their water intake was restricted to 16 h to prevent anti-diuresis. Rats in both groups in which labor occurred within 24 h after the experiment were sacrificed and the tissue sample of the muscles were taken. In each rat, the m. coccygeus, m. iliocaudalis and m. pubocaudalis muscles located on the left side were dissected. The following bony landmarks were used to identify the origin/insertion points of m. coccygeus, m. iliocaudalis and m. pubocaudalis, respectively: pubic bone/insertion caudal (Ca) 1–2 vertebrae, ilium/Ca5-Ca6 vertebra and pubic bone/Ca3-Ca4 vertebrae (Figure 1) [18].

### 2.1. Examination of Pelvic Floor Muscles

The dissected PFMs were stored in tubes containing 10% formaldehyde for 3–5 d for fixation. To minimize the risk of bias in the study, a pathologist blinded to the study protocol evaluated the tissue samples. After 1 d of fixation and follow-up, the tissues were embedded in paraffin by using a vacuum tissue processor (Leica TP 1050). Subsequently, 4 µm-thick sections were cut from the tissues using a microtome (Leica 2125). An automatic staining device (Leica 5020) was used for hematoxylin and eosin staining.

After the pathological examination of the sections, the images were uploaded to a Leica-Aperio CS2 scanner for precise measurement of the tissues. The digital slides were examined at large magnification, and the vertical and horizontal dimensions of three randomly transverse cut muscle cells were measured and recorded using the pathology slide viewing software program (Aperio ImageScape) (Figure 2 and Figure 3).

### 2.2. Preparation of Oxytocin Solution

2.5 mU oxytocin was prepared by mixing 10 units of oxytocin (a) with 10 mL of saline [36].

### 2.3. Statistical Analysis

Data were analyzed using IBM SPSS V23. Comparisons of the vertical and horizontal dimensions of the muscles’ mean values between groups were analyzed using a one-way analysis of variance; multiple comparisons of the groups were analyzed using Tukey’s HSD. The paired sample *t*-test was used to determine the statistical significance of mean differences between experiment and control groups. The relationship between the rat weights and the vertical and horizontal aspect of the muscle measurements were examined by Pearson’s correlation. The results are presented as mean ± standard deviation. *p* values of <0.05 were considered significant.

## 3. Results

### 3.1. Mean Values of Sections from the Rats

The mean values of the body weights in group I were lower than those in the other groups (Table 1) (*p* < 0.001).

When the mean body weights of the rats and the vertical and horizontal dimensions of the muscles were compared, there was a strong negative correlation between the vertical dimension of the m. pubocaudalis muscle and body weight in group I (*r* = −0.813; *p* = 0.014) and a strong negative correlation between the vertical dimension of the m. coccygeus muscle and body weight in group II (*r* = −0.817; *p* = 0.013). There was also a strong negative correlation between the horizontal dimensions of the m. coccygeus muscle and body weight in group II (*r* = −0.757; *p* = 0.030) and a strong negative correlation (*r* = −0.734; *p* = 0.038) between the vertical dimension of the m. coccygeus muscle and body weight in group III (Table 2) (*p* > 0.05). In group IV, no correlation was found between the body weight and pelvic floor muscle dimensions.

Between-group comparisons of the vertical and horizontal dimensions of the m. iliocaudalis muscle revealed no between-group differences in the average values in any of the groups (Table 3) (*p* = 0.831 and *p* = 0.266, respectively).

There was no difference in the mean value of the vertical dimension of the m. pubocaudalis muscle between groups (*p* = 0.604). When the mean values of the horizontal dimensions of the m. pubocaudalis muscle were compared in the various groups, the mean value in group IV (50.1 ± 5.4 µm) was higher than that in the other groups (Table 4) (*p* < 0.001).

There was no between-group difference in the average values of the vertical dimensions of the m. coccygeus muscle (*p* = 0.099). The mean values of the horizontal dimensions of the m. coccygeus muscle differed between the groups, with the mean value in group III (49.5 ± 10.9 µm) higher than that in group I (Table 5) (*p* = 0.009).

Figure 2 shows the image obtained from the m. pubocaudalis (PC) muscle of the rat in the oxytocin group. Accordingly, (H&E, X200) striated muscle cells are observed.

Figure 3 shows the image obtained from the m. coccygeus (C) muscle of the rat in the saline solution group. Accordingly, (H&E, X200) striated muscle cells are observed.

### 3.2. Observational Findings

In addition to statistical and light microscopic findings, observational findings were also recorded. Based on the observational findings, the majority of the rats in group III and group IV exhibited a tendency towards cannibalism after birth. However, cannibalism was not observed in group II. In addition, the number of stillbirths in groups III and IV was higher than that in the other groups, and no stillbirths were recorded in group II.

## 4. Discussion

The striated pelvic floor muscles (PFMs), comprise the coccygeus and the muscles of the levator ani complex [2]. However, in many studies in the literature, PFM and levator ani muscle are used interchangeably and the C muscle is not included [37]. It is known that striated muscles are most susceptible to injury when they are forcibly extended [38]. The level of injury to the muscle is proportional to the impact the muscle is exposed to during stretching [38]. Modeling and imaging studies estimate that vaginal delivery causes > 300% strain in the PFMs [2]. The strain at this level exceeds the physiological limit of striated muscles and causes PFM trauma [2]. Therefore, vaginal delivery is one of the important reasons of PFM trauma [37]. 

Synthetic oxytocin is commonly used for labor induction, augmentation and postpartum care in modern obstetrics [39,40]. Use of oxytocin itself was reported to be a risk factor for perineal damage during vaginal delivery, yet research on the implications beyond labor of maternal exposure to perinatal synthetic oxytocin is rare and in particular, there are very few studies examining the effect of oxytocin induction on pelvic floor muscles [33,35,39,40]. Therefore, the present study was conducted to investigate the effects of oxytocin induction on pelvic floor muscles in pregnant rats.

Architectural studies require isolation of the whole muscle and this type of research is not feasible in women [2]. In rats, the location of coccygeus, iliocaudalis, and pubocaudalis muscle are anatomically similar to human PFMs [2]. Therefore, we used rats (virgin-group I, spontaneous vaginal delivery-group II, saline control-group III, and oxytocin-group IV) in our study. Our data showed that there were no between group differences in m. iliocaudalis muscle vertical and horizontal dimensions, whereas m. pubocaudalis muscle horizontal dimension was significantly larger in group IV, and m. coccygeus muscle horizontal dimension was significantly larger in group III.

In our study, it was determined that there was no difference between the groups in iliocaudalis muscle vertical and horizontal dimensions. Alperin et al. [37] examined the thickness level in the pelvic floor muscles due to the tension and stress occurring in the pelvic floor muscles during pregnancy and they found that there was no change in the m. iliocaudalis muscle. On the other hand, according to Shi et al. [41], a study conducted by magnetic resonance method indicates that the change has occurred in the m. iliocaudalis muscle. Alperin et al. [2] showed that m. iliocaudalis muscle size was significantly enlarged in the postpartum group compared to the virgin group (14.8 ± 0.24 mm in the virgin group and 18.7 ± 0.61 mm in the postpartum group, *p* < 0.001). We think that this difference may be due to the difference in research methodologies.

In our study, when the difference between the groups in terms of horizontal size averages of the m. pubocaudalis muscle was examined, the mean value (50.1 ± 5.4 µm) obtained in the oxytocin group was found to be statistically significantly higher than the other groups (*p* < 0.001). Shi et al. [41], in a study they have done with a magnetic resonance imaging method, concluded that the m. pubocaudalis muscle of the levator ani muscle was the most damaged in primipara women. Likewise, Lien et al. [38] and Parente et al. [42] determined that the m. pubocaudalis muscle was damaged the most. These similar results may have resulted from the location of the m. pubocaudalis muscle in both humans and experimental animals.

In our study, when the mean values of the vertical and horizontal dimensions of the m. coccygeus muscle were compared according to the groups, the horizontal size of the m. coccygeus muscle differed according to the groups, and it was seen that the average value obtained in the saline group was higher than the virgin group (*p* = 0.009). Alperin et al. [37], in a study in which they examined the thickness level of the pelvic floor muscles due to the tension and stress in the pelvic floor muscles during pregnancy, stated that the highest level of stiffness occurred in the m. coccygeus muscle (52%) compared to other muscles. Similarly, Catanzarite et al. [43] stated that the most tension occurs in the m. coccygeus muscle compared to other pelvic floor muscles during vaginal delivery. Lindo et al. [17] and Bracken et al. [19] stated in their study on monkeys that the volume of the m. coccygeus muscle increased due to pregnancy and vaginal delivery. On the other hand, Alperin et al. [2] stated in a study they conducted that the m. coccygeus muscle mass did not change due to pregnancy and birth. We think that this difference may be due to the difference in research methodologies.

Evidence suggests that rats and mice tend to eat their young more than other species. Rats are prone to cannibalism, especially in the presence of stressful environmental stimuli such as high frequency noise, frequent cage changing, excessive cage movement, water or food restriction [44]. Observational studies have found an association between oxytocin use and adverse outcomes for newborns [32]. During our study, we also paid attention to the observational findings. In this study, the majority of the rats in group III and group IV showed a tendency towards cannibalism after birth, whereas this was not observed in group II. In addition, the number of stillbirths in groups III and IV was higher than that in the other groups, and no stillbirths were recorded in group II. Any intervention in spontaneous vaginal delivery affects the physiology of birth [3]. Ozturk and Sayiner reported that environmental changes such as social stress, immobilization, water and food restriction, light, darkness and crowding, diversely affect the physiology of spontaneous vaginal delivery [45]. These findings may be attributed to intervening in the normal life cycle of the rats.

Although there are many studies related to uterine rupture [9,10] and pelvic floor dysfunction due to pregnancy and delivery in the clinic, we could not find any study in the literature that histologically demonstrated the effect of oxytocin hormone on pelvic floor muscles. To our knowledge, this is the first study in the literature demonstrating the effects of oxytocin induction on pelvic floor muscle in rats. In our study, we observed that there was no significant difference in vertical and horizontal dimensions between the groups for m. iliocaudalis muscles, but we found that the mean value of the horizontal dimension of the m. pubocaudalis muscle was significantly larger in group IV compared to groups I, II and III. In addition, we found that the mean value of the horizontal dimension of the m. coccygeus muscle in group III was significantly larger when compared to the mean value of the horizontal dimension of the m.coccygeus muscle in group I. In our study, besides the statistical data, it was observed that most of the rats in the saline solution and oxytocin groups showed a tendency to cannibalism after birth. However, such a situation was not encountered in the spontaneous vaginal delivery group. In addition, rats in the saline solution and oxytocin groups showed a tendency to give birth to stillborn puppies, while stillborn puppies were not detected in the spontaneous vaginal delivery group.

In conclusion, we can say that oxytocin administration does not have a direct harmful effect on the pelvic floor muscles, and that the pregnancy process and labor may have made histological changes on these muscles. In addition, we think that the birth process and the birth environment should not subjected to intervention unless necessary.

## Figures and Tables

**Figure 1 medicina-59-00234-f001:**
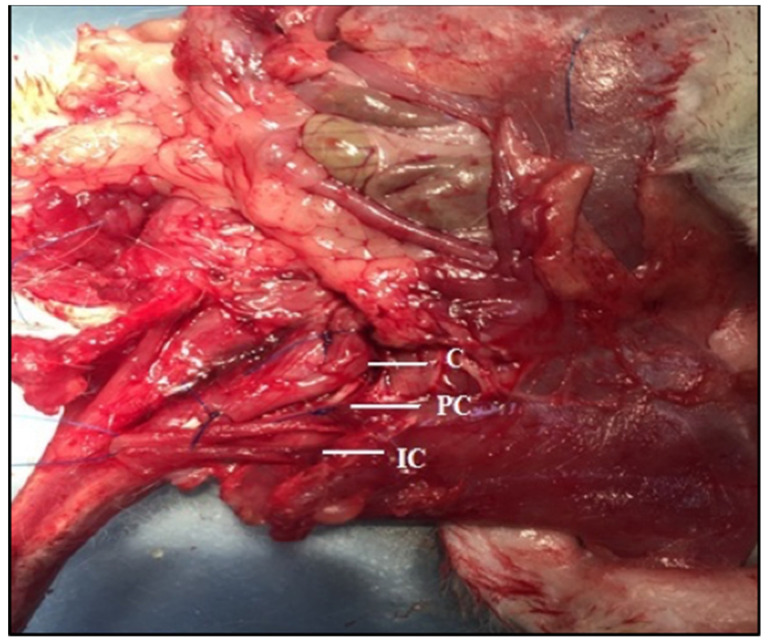
Rat left m. iliocaudalis (IC), m. pubocaudalis (PC) and m. coccygeus (C) muscles.

**Figure 2 medicina-59-00234-f002:**
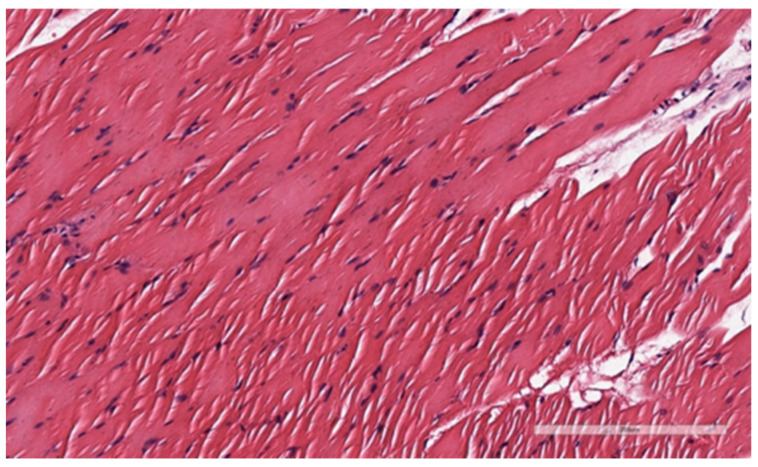
Oxytocin group m. pubocaudalis (PC) muscle.

**Figure 3 medicina-59-00234-f003:**
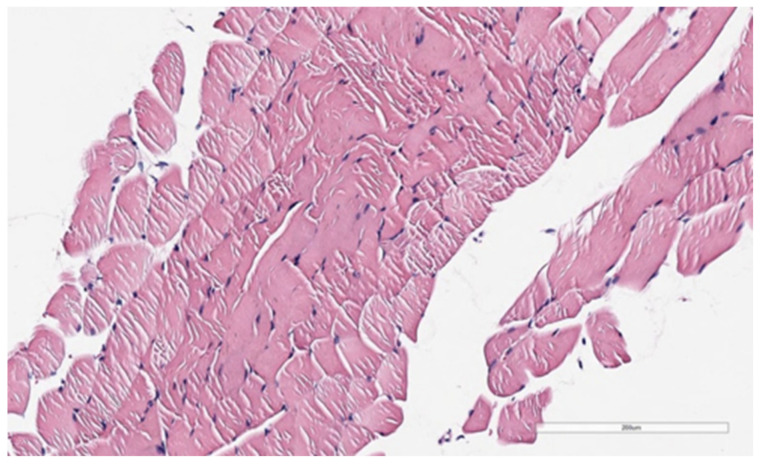
Saline solution group m. coccygeus (C) muscle.

**Table 1 medicina-59-00234-t001:** Average values of body weight of rats by groups.

Groups (n = 8)	Body Weight (g)
Group I	169.3 ± 0.9 (a)
Group II	271 ± 1.1 (b)
Group III	267.5 ± 30 (b)
Group IV	276 ± 22 (b)
Test Statistics	F = 60.481
p	<0.001

Group I: Virgin group, Group II: Spontaneous vaginal delivery group, Group III: Saline solution group, Group IV: Oxytocin group, F: One-way analysis of variance, a-b: There is no difference between groups with the same letter in the corresponding column.

**Table 2 medicina-59-00234-t002:** Comparison of vertical and horizontal dimensions of muscle body weights.

Muscles		Group I	Group II	Group III	Group IV
m.iliocaudalis vertical	r	−0.287	0.329	−0.352	0.051
	p	0.491	0.427	0.393	0.904
m.iliocaudalis horizontal	r	0.358	0.377	−0.669	0.632
	p	0.384	0.357	0.070	0.093
m.pubocaudalis vertical	r	−0.813	0.261	0.115	−0.352
	p	0.014	0.533	0.787	0.392
m.pubocaudalis horizontal	r	−0.271	−0.217	0.500	0.558
	p	0.516	0.605	0.207	0.150
m.coccygeus vertical	r	0.111	−0.817	−0.734	0.303
	p	0.794	0.013	0.038	0.466
m.coccygeus horizontal	r	0.510	−0.757	0.499	−0.432
	p	0.196	0.030	0.208	0.285

Group I: Virgin group, Group II: Spontaneous vaginal delivery group, Group III: Saline solution group, Group IV: Oxytocin group.

**Table 3 medicina-59-00234-t003:** Comparison of the mean values of the vertical and horizontal dimensions of the m. iliocaudalis muscle according to the groups.

Groups (n = 8)	Vertical (µm)	Horizontal (µm)	Test Statistics
Group I	60.1 ± 15.7	36.7 ± 8.2	t = 3.999
Group II	63.4 ± 7.1	36.7 ± 4.8	t = 10.162
Group III	63.4 ± 11.1	42.2 ± 4.7	t = 5.736
Group IV	65.1 ± 7.2	40.1 ± 7.4	t = 9.332
Test Statistics	F=0.292	F = 1.392	
p	0.831	0.266	

Group I: Virgin group, Group II: Spontaneous vaginal delivery group, Group III: Saline solution group, Group IV: Oxytocin group, F: One-way analysis of variance, t: Dependent samples *t*-test.

**Table 4 medicina-59-00234-t004:** Comparison of the mean values of the vertical and horizontal dimensions of the m. pubocaudalis muscle according to the groups.

Groups (n = 8)	Vertical (µm)	Horizontal (µm)	Test Statistics
Group I	67.2 ± 15.7	35.7 ± 5.2 (a)	t = 5.853
Group II	73.8 ± 10.0	36.3 ± 2.8 (a)	t = 11.901
Group III	76.7 ± 15.3	37.7 ± 4.3 (a)	t = 6.514
Group IV	74.8 ± 16.9	50.1 ± 5.4 (b)	t = 4.255
Test Statistics	F = 0.626	F= 18.125	
p	0.604	<0.001	

Group I: Virgin group, Group II: Spontaneous vaginal delivery group, Group III: Saline solution group, Group IV: Oxytocin group, F: One-way analysis of variance, t: Dependent samples *t*-test, a-b: There is no difference between groups with the same letter in the corresponding column.

**Table 5 medicina-59-00234-t005:** Comparison of the mean values of the vertical and horizontal dimensions of the m. coccygeus muscle by groups.

Groups (n = 8)	Vertical (µm)	Horizontal (µm)	Test Statistics
Group I	64.7 ± 12.3	37.6 ± 5.6 (a)	t = 6.165
Group II	88.7 ± 29.5	40.3 ± 6.5 (ab)	t = 5.756
Group III	70.7 ± 5.5	49.5 ± 10.9 (b)	t = 8.466
Group IV	79.6 ± 22.0	46.7 ± 4.5 (ab)	t = 4.482
Test Statistics	F = 2.302	F = 4.642	
p	0.099	0.009	

Group I: Virgin group, Group II: Spontaneous vaginal delivery group, Group III: Saline solution group, Group IV: Oxytocin group, F: One-way analysis of variance, t: Dependent samples *t*-test, a-b: There is no difference between groups with the same letter in the corresponding column.

## Data Availability

Not applicable.

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
