# Peer review of "Effects of the Oxytocin Hormone on Pelvic Floor Muscles in Pregnant Rats"

_medicina, 2023, doi:10.3390/medicina59020234_

Round 1
Reviewer 1 Report
-The work is certainly original, but I would like to understand what is the ultimate goal?
What implications can it have from a clinical point of view?
I would like a paragraph of conclusions where the objective of the study and the conclusions are clearly highlighted.
-This sentence is not clear:
“Also, our observational findings show that in the clinical setting, all steps should be taken to support a natural labor process (i.e. spontaneous vaginal delivery) to ensure that the physiology of labor is not disturbed.”
Please explain it better
- Moreover, for further information I suggest to cite these articles:
PMID: 36129411.
PMCID: PMC9317678
PMCID: PMC9317678
Author Response
Dear referee,
The revisions you requested are attached.
Kind regards.

Reviewer 2 Report
In my opinion, the document is an interesting analysis of the function of oxytocy in an animal model. After reading the document, I believe that the article entitled Effects of the Oxytocin Hormone on Pelvic Floor Muscles in
Pregnant Rats is well prepared and suitable for publication in Medicina
Author Response
Dear referee,
Thanks for your feedback on our work.
Kind regards.
Reviewer 3 Report
This is an interesting manuscript about the effects of the Oxytocin Hormone and induction of labor on Pelvic Floor Muscles(PFMs) in Pregnant Wistar Rats. The authors analysed the characteristics of PFMs for 32 female Wistar Albino divided into four groups Group I: virgin rats which has gave no delivery before. Group II: pregnant rats which gave spontaneous vaginal delivery. Group III: Delivery was induced with saline solution intravenously (iv) on the 21 d of pregnancy at 10 min intervals for 8 h and saved as a control group. Group IV: Delivery was induced with oxytocin solution iv on the 21 d of pregnancy at 10 min intervals for 8 h and saved as an experimental group. Group I (virgin group) were anesthetized under deep anesthesia, and the PFMs were dissected at the age of 3 months. Group II,III and IV were sacrificed after labor and vaginal delivery (spontaneous, induced without and with Oxytocin) .
I suggest the following in order to better understand :
- In the Introduction the authors must explain what animal models that are suitable for studying parturition-induced pelvic floor disorders and what are the pelvic floor muscles for humans and for rats .
- In Material and Methods : I suggest to describe the dissection and identification of muscles
- The authors related cannibalism for Groups III and IV ? it is a particularity ? or in animals is normal ?
- Discussion : the authors made comments on human pelvic floor muscles , it is important to do correlation with animals beyond the location of the muscles
- Conclusions: vaginal birth or pregnancy or both affects PFMs he authors must explain .
Author Response

(The authors gave the same response as above.)

Round 2
Reviewer 1 Report
Now the manuscript for me is ok.